# Significance of acPWV for Survival of Hemodialysis Patients

**DOI:** 10.3390/medicina56090435

**Published:** 2020-08-28

**Authors:** Marko Petrovic, Marko Baralic, Voin Brkovic, Aleksandra Arsenovic, Vesna Stojanov, Natasa Lalic, Dejana Stanisavljevic, Aleksandar Jankovic, Nenad Radivojevic, Svetlana Pejanovic, Ivko Maric, Visnja Lezaic

**Affiliations:** 1Clinical Centre of Serbia, Department of Nephrology, 11000 Belgrade, Serbia; markofbkbgd@gmail.com (M.P.); baralicmarko@yahoo.com (M.B.); voin.brkovic@gmail.com (V.B.); svetlanapejanovic@yahoo.com (S.P.); 2School of Medicine, University of Belgrade, 11000 Belgrade, Serbia; stojanovves@eunet.rs; 3Special Hospital for Endemic Nephropathy, 11555 Lazarevac, Serbia; aleksandra.arsenovic@yahoo.com (A.A.); nefropatija@eunet.rs (I.M.); 4Clinical Centre of Serbia, Department of Cardiology, 11000 Belgrade, Serbia; kardiologija@klinicki-centar.rs; 5Clinical Centre of Serbia, Centre for Medical Biochemistry, 11000 Belgrade, Serbia; natasalalic@hotmail.com; 6Institute for Medical Statistics, School of Medicine, University of Belgrade, 11000 Belgrade, Serbia; sdejana8@yahoo.com; 7Clinical Department for Renal Diseases, Zvezdara University Medical Center, 11000 Belgrade, Serbia; sashajan223@gmail.com

**Keywords:** prevalent hemodialysis patients, arterial stiffness, biomarkers, cardiovascular assessment, outcome

## Abstract

Background and Objectives: Abnormal arterial stiffness (AS) is a major complication in end-stage kidney disease (ESKD) patients treated by dialysis. Our study aimed to determine the significance of AS for survival of prevalent dialysis patients, as well as its association with cardiovascular parameters or vascular calcification promoters/inhibitors or both and AS. Materials and Methods: The study involved 80 adult hemodialysis patients. Besides standard laboratory analyses, we also determined promoters and inhibitors of vascular calcification (bone biomarkers): serum levels of fibroblast growth factor 23 (FGF23), soluble Klotho, intact parathormone (iPTH), 1,25-dihydroxyvitamin D3, osteoprotegerin, sclerostin, AS measured as ankle carotid pulse wave velocity (acPWV), Ankle Brachial Index (ABI), and vascular calcification (VC) score. Patients were monitored for up to 28 months. According to the median acPWV value, we divided patients into a group with acPWV ≤ 8.8 m/s, and a group with acPWV > 8.8 m/s, and the two groups were compared. Results: Values for bone biomarkers were similar in both groups. Mean arterial blood pressure (MAP), central systolic and diastolic brachial blood pressure, heart rate, and pulse pressure were higher in the group with acPWV > 8.8 m/s than in the group with acPWV ≤ 8.8 m/s. The mortality was higher for patients with acPWV > 8.8 m/s at any given time over 28 months of follow-up. In multivariable analysis, predictors of higher acPWV were age >60.5, higher pulse rate, and higher central systolic or brachial diastolic blood pressure. Conclusions: According to our results, we advise the measurement of acPWV preferentially in younger dialysis patients for prognosis, as well as intervention planning before the development of irreversible changes in blood vessels. In addition, measuring central systolic blood pressure seems to be useful for monitoring AS in prevalent hemodialysis patients.

## 1. Introduction

Abnormal vascular remodeling (VR) is a major complication of chronic kidney disease (CKD) and end-stage kidney disease (ESKD) [1]. The arterial remodeling in these cases includes atherosclerosis of medium-sized conduit arteries, and arteriosclerosis characterized by increased vascular calcifications (VC) and arterial stiffness (AS) of the aorta and large arteries [1]. The reduction in arterial distensiblity greatly affects blood pressure regulation, i.e., raises systolic and reduces diastolic blood pressure, which then increases cardiac afterload and compromises perfusion of the coronary arteries during diastole [1]. These changes are even more profound if coronary plaque is present [2]. Subsequent left ventricular remodeling and hypertrophy elevates the risk of myocardial infarction and heart failure, and ultimately favors increased cardiovascular mortality [1,2,3,4]. On the other hand, the presence and progression of extensive VC and AS have been shown to be independently associated with mortality in chronic dialysis patients [4,5,6,7,8]. However, the addition of AS measured with ankle carotid pulse wave velocity (acPWV) to standard clinical risk scores only modestly improved CV risk reclassification of ESKD patients’ mortality [9].

VR results from a complex interaction between structural and functional changes in the vessel wall under the influence of traditional and nontraditional cardiovascular risk factors, especially oxidative stress, endothelial dysfunction, mineral metabolism disorders, and renal bone disease [10,11,12,13]. At the molecular level, several factors and mechanisms underlying the development of VR in patients with CKD and ESRD have been put forward. Extensive studies on the occurrence of VC show that systemic or local inhibitory factors, such as matrix Gla protein, fetuin-A, osteopontin, osteoprotegerin, and pyrophosphate, are most likely overwhelmed in CKD patients by promoters (calcium and phosphate abnormalities, extreme serum PTH levels, excess administration of calcium salts, inflammation, malnutrition, and oxidative stress) that induce vascular smooth muscle cell damage and death. In addition, fibroblast growth factor 23 (FGF23) and its co-receptor, klotho, have emerged as key regulators of mineral homeostasis. All these lead to an undesirable imbalance favoring excessive calcification [3]. As VC is associated with AS, the same markers were studied for AS pathophysiology [8,14,15,16]. While some of the mentioned substances have been investigated as an underlying mechanism of AS in animal models, observational human studies are not consistent. Thus, a positive relationship between increased osteoprotegerin and AS development in CKD patients was reported [17], but others did not confirm this relationship [18,19].

Bearing in mind all these controversies, we conducted the present study with the aim to determine (a) the significance of AS for survival of prevalent dialysis patients, and (b) potential independent association between cardiovascular parameters or VC promoters/inhibitors or both and AS.

## 2. Materials and Methods

### 2.1. Study Groups

The study involved 80 non-diabetic adults on hemodialysis (HD) selected from the pool of patients treated with chronic hemodialysis in two nephrology departments (Clinical Centre of Serbia and Lazarevac). Exclusion criteria were (1) hemodialysis duration of less than 6 months; (2) patients disagreed to participate in the study, which was approved by the institutional review board; (3) acute CV complications during the 6 months preceding entry into the study; (4) hemodynamically significant lower extremity artery occlusive disease; (5) atrial fibrillation at the time of pulse wave velocity (PWV) measurement; and (6) uncontrolled blood pressure.

The participants were monitored from October 2015 until death or February 2020. The Ethics Committee of the Clinical Centre of Serbia evaluated and approved the study protocol (decision no. 1690/21, 9 June 2015), and all patients provided written informed consent. Biochemical, calcification, and vascular assessment were performed after signing the informed consent.

Standard bicarbonate hemodialysis sessions lasted 12 h weekly. Dialysate calcium (dCa) was individualized to meet the specific requirements of each patient by optimizing management of serum Ca, phosphate, parathyroid hormone, and alkaline phosphatase levels and could be altered. Dialysate Ca at 1.5 mmol/L was commonly used; dCa of 1.25 mmol/L was applied to permit the use of vitamin D supplements and Ca-based phosphate binders in the setting of biochemically suspected adynamic bone disease; dCa higher than 1.75 mmol/L was employed for suppression of hyperparathyroidism, taking into account side effects. Management of renal osteodystrophy was adjusted on the basis of drug availability. Before the start of this cross-sectional study, the patients had received paricalcitol for 12 to 22 months, except for two individuals who had taken the drug for 4 months and 80 months, respectively. In addition, a smaller number of patients were prescribed sevelamer for up to 12 months. Cinacalcet was given to only a few patients for up to 6 months. During the study and in the following period, neither sevelamer nor cinacalcet were used. 

Additional variables of interest from the patients’ records were demographic (age, gender); underlying kidney disease; dialysis duration; systolic and diastolic blood pressures recorded during dialysis; and previous history of cerebrovascular disease and cardiovascular diseases, including coronary artery disease, congestive heart failure, peripheral vascular disease, parathyroidectomy, and medicaments (antihypertensives, phosphate binders, vitamin D, erythropoietin stimulating agents (ESAs)).

### 2.2. Biochemical Analyses

Data were recorded for serum urea, creatinine, uric acid, total protein, C reactive protein (CRP), lipid profile, phosphate, calcium, alkaline phosphatase, hematological parameters, iron status, and intact parathormone (iPTH). Mean standard weekly Kt/V and urea reduction ratio (URR) were calculated. We also measured serum levels of 1,25-vitamin D3, fibroblast growth factor 23 (FGF23) and soluble Klotho, osteoprotegerin, fetuin A, and sclerostin.

The routine biochemical parameters were determined using standard techniques, while stimulators or inhibitors of VC were analyzed in the same laboratory in the Clinical Centre of Serbia. Serum iPTH was measured by immunoradiometric assay (ELSA-PTH, CIS Bio International), with normal values being 11 to 62 pg/mL. Serum levels of FGF23 were assayed with a commercially available kit (Cusabio, Houston, TX, USA) using an ETI-max 3000 (Dia-Sorin, Saluggia, Italy). According to the manufacturer, the assay has a measurement range of 3.12–200 pg/mL with a lower limit of detection of 0.78 pg/mL. Intra-assay precision was <8%, and inter-assay precision was <10%. Serum levels of soluble Klotho were determined using a commercially available kit (Cusabio, Houston, TX, USA) and ETI-max 3000 (Dia-Sorin, Saluggia, Italy). According to the manufacturer, the assay measurement range is 156–10 ng/mL, with a lower limit of detection of 0.039 ng/mL. Intra-assay precision was <8%, and inter-assay precision was <10%. A commercial chemiluminescent immunoassay (Diasorin S.p.A., Saluggia, Italy) was employed to determine serum 1,25(OH)2D on a LIAISON Analyzer (Diasorin S.p.A., Italy). The measuring range is 7.6–147.8 ng/mL, limit of quantification is 3.5 ng/mL, and the intra- and inter-assay coefficients of variation are 2.5% and 6.5%, respectively. Serum levels of osteoprotegerin were determined using a commercially available kit (Biomedica, Vienna, Austria). According to the manufacturer, the assay range is 0–20 pmol/L (0–400 pg/mL), and the intra- and inter-assay coefficients of variation are ≤5% and ≤3%, respectively. Serum levels of sclerostin were determined using a commercially available kit (Elabscience, Houston, TX, USA). According to the manufacturer, the detection assay range is 62.4–4000 pg/mL, and the intra- and inter-assay coefficients of variation are <10%.

### 2.3. Calcification Assessment

Vascular calcification in the iliac, femoral, radial, and digital arteries in plain radiographic films of the pelvis and hands were evaluated by one person. A simple VC score was calculated as described by Adragao et al. [20]. In brief, the pelvis radiographic films were divided into four sections by two imaginary lines: a horizontal line over the upper limit of both femoral heads and a median vertical line over the vertebral column. The films of each hand were divided by a horizontal line over the upper limit of the metacarpal bones. The presence of linear calcifications in each section was counted as 1, and its absence as 0. The final score was the sum of all the sections, ranging from 0 to 8. Vascular calcification was graded as follows: 0 = no calcification, 1–3 = mild calcification, >4 = severe calcification.

### 2.4. Brachial Blood Pressure

Brachial blood pressure (BP) was measured in all patients in the non-fistula arm. Blood pressure was measured three times at 1-min intervals according to the European Society of Hypertension/Cardiology guidelines [21]. A validated automated oscillometric device was used for all measurements with a medium- or a large-size cuff, according to the participant’s arm circumference. BP was measured during hemodialysis every hour from the beginning of hemodialysis, and the value for statistical processing was expressed as the mean value of systolic and diastolic BP.

### 2.5. Vascular Assessments

A Complior SP system (Artech Medical, Pantin, France) was used simultaneously to assess the central systolic blood pressure (sSBP) and ankle carotid pulse wave velocity (acPWV) by two trained investigators. acPWV was measured 1 day between mid-week hemodialysis, utilizing two sensors (one carotid and one femoral) simultaneously to determine the velocity of the pulse wave in relation to the distance between the femoral artery and the suprasternal notch. Two measurements were taken, and the mean value was calculated. To determine the inter-observer variability of acPWV measurements, two experienced investigators independently analyzed 20 randomly selected patients. Interobserver variability was 7%.

The Ankle Brachial Index (ABI) was determined 10–30 min before HD using an ABI-form device that automatically and simultaneously measured blood pressure in both arms and ankles with a Doppler device [22]. Briefly, with the patient in a supine position, occlusion and monitoring cuffs were placed tightly around the upper arms without blood access and on both lower extremities. The ABI was calculated as the ratio of the ankle systolic BP divided by the arm systolic BP. ABI measurements were made twice for each patient. Low ABI (<0.9) identifies obstructive artery disease, while high ABI (>1.3) is caused by stiff non-compressible distal arteries, probably due to distal arterial calcification [23].

### 2.6. Statistical Analysis

Continuous variables are presented as the mean (standard deviation (SD)) for normally distributed variables and as the median (interquartile range (IQR)) for non-normally distributed variables. Chi-square tests, ANOVA, or Kruskal–Wallis one-way ANOVA were used to examine differences in various baseline variables between the groups of patients. Patients were divided into two groups according to median acPWV value: ac PWV ≤ 8.8 m/s, and ac PWVT2 > 8.8 m/s, and were analyzed as a categorical variable. Univariate and multivariate Cox proportional hazard models were employed for determination of hazard ratios (HR) that variables have for all-cause and CV mortality. All variables that were statistically significant in the univariate analysis were included in the multivariate analysis. Model 1 was adjusted for age, sex, and brachial BP, while Model 2 was adjusted for age, sex, central systolic blood pressure (cSBP), and brachial systolic blood pressure (SBP). The Kaplan–Meier method was used for survival plots that were compared by log-rank tests. Receiver operating characteristics (ROC) curve analysis was performed using age as a continuous variable and acPWV as outcome in order to determine the most sensitive and specific patient age cut-offs for acPWV over 8.8 m/s. Statistical analysis was performed with SPSS 21.0 (SPSS, Inc, Chicago, IL, USA). The probability (*p*-value) ≤ 0.05 was considered statistically significant.

## 3. Results

### 3.1. Basal Clinical Data

The baseline characteristics, presence of co-morbidities, and treatment of the participants in two acPWV groups are shown in Table 1. Patients with PWV > 8.8 m/s were older than patients with acPWV ≤ 8.8 m/s. Eight patients (30.5%) with acPWV ≤ 8.8 m/s and 26 patients with acPWV > 8.8 m/s (72%) were older than 60.5 years (the oldest patient was 83 years of age in the latter group. The ROC curve showed Area Under the Curve (AUC) 0.762 (*p* < 0.001) and revealed that 60.5 years was the most sensitive and specific (68.6% and 77.1%, respectively) cut-off age value for the studied patients. There were no significant differences in dialysis vintage prior to inclusion in the present study, as was the case for body mass index, presence of hypertension, cardiovascular disease (CVD) and cerebrovascular insult (CVI) co-morbidities, treatment with ESA, phosphate binders, and antihypertensives between the two groups. The distribution of primary causes of ESRD were similar, but common renal calculosis in the group with acPWV ≤ 8.8 m/s and common Balkan endemic nephropathy in the group with acPWV > 8.8 m/s were found.

### 3.2. Basal Clinical Findings, Blood Vessel Parameters, and Laboratory Data of Patients from Different acPWV Groups

Data on blood vessel function and morphology and blood pressure parameters for the two groups are presented in Table 2. Vascular calcifications were found in 51 (70.8%) subjects, while an Adragao score > 4 was present in 17 (23.6%) of them. Among the individuals with the highest Adragao score of 8, one was in the acPWV ≤ 8.8 m/s group and three were in the acPWV > 8.8 m/s group. Mean arterial blood pressure (MAP), cSBP, systolic and diastolic brachial blood pressure, heart rate, and pulse pressure were higher in the group with acPWV > 8.8 m/s than in the group with acPWV ≤ 8.8 m/s.

Basal laboratory analyses are shown in Table 3. Serum urea and alkaline phosphatase were significantly higher in the acPWV > 8.8 m/s group. Values for promoters and inhibitors of vascular calcifications iPTH, FGF23, Klotho, magnesium, osteoprotegerin, and sclerostin were similar in both groups.

### 3.3. Patient Mortality and Predictors

During the follow-up period, 10 patients died due to heart failure, acute myocardial infarction, aorto-coronary bypass surgery, peripheral vascular disease with ulceration (four patients), cerebrovascular insult (one patient), systemic infection (two patients), colorectal cancer, and hepatocellular carcinoma and decompensated cirrhosis following hepatitis B virus infection (one case each).

Patient survival curves for the acPWV median groups are presented in Figure 1. Kaplan–Meier analysis revealed an increased rate of mortality with increase of acPWV > 8.8. Mean survival was 322.67 ± 19.05 months for patients with acPWV ≤ 8.8, and 205.78 ± 17.78 months for patients with acPWV > 8.8, with the difference being significant (log rank 9.633, *p* = 0.002).

Age, arterial hypertension, median acPWV, and Klotho were selected in univariate Cox regression analysis as predictors of mortality. However, having acPWV > 8.8 m/s was selected as an independent predictor for a fatal outcome in multivariate analysis (Table 4). The probability of dying was higher for patients with acPWV > 8.8 m/s at any given time over 28 months of follow-up.

The independent variables significantly associated with acPWV selected by logistic regression analysis are presented in Table 5 as two models. A higher acPWV was associated with older age, higher pulse rate, and higher central systolic or brachial diastolic blood pressure.

## 4. Discussion

The present study showed that acPWV is an independent risk factor for prevalent hemodialysis patients’ all-cause mortality, and the cut-off value for acPWV is above 8.8 m/s. In addition, increasing acPWV was positively associated with older age, higher pulse rate, and higher central systolic and brachial diastolic blood pressure.

Our results are in accordance with previous studies, which have shown that acPWV contributes to increased cardiovascular morbidity and mortality in patients with CKD 2–5 and those treated by hemo- and peritoneal dialysis [13,24,25,26,27]. Moreover, epidemiological studies have highlighted high AS as a risk factor for development of cardiovascular disease and mortality in different non-CKD populations, including those with no previous cardiovascular disease [28,29] or with uncomplicated essential hypertension [30]. Moreover, acPWV improves prediction of CV events in patients with CVD and stable coronary artery disease undergoing Percutaneous Coronary Intervention instead(PCI) and may identify high-risk populations who may benefit from aggressive CV risk factor management [31,32,33]. Okhuma et al. found that acPWV could enhance prediction of the risk for development of CVD over that of the Framingham risk score, which is based on traditional cardiovascular risk factors [29]. All this led the European Society of Hypertension and the European Society of Cardiology to recommend determination of acPWV as the “gold standard” to estimate central arterial changes and a reliable predictor of increased cardiovascular risk [34,35].

The reference values for acPWV were assessed from 11,092 European subjects without overt CV disease, diabetes, treated hypertension, or dyslipidemia [36]. It was found that acPWV is higher in males and rises steadily with advancing age and increasing blood pressure category. Furthermore, in this population, mean and median reference values were provided for each age and blood pressure category [36]. Different acPWV cut-off values for survival of dialysis patients have been reported, from 16.6 m/s [13] up to 8.5 m/s [5,24,25,26,37,38]. There are a few possible explanations. Dialysis patients have premature arterial aging as well as premature AS. Moreover, it cannot be neglected that the authors obtained different cut-off values in different studies by dividing patients into groups in terciles and quartiles. As there is no consensus about the cut-off acPWV values relevant to predict mortality of dialysis patients, we attempted a somewhat different approach to this question, i.e., we divided our patients into two subgroups: those with median acPWV equal or less than 8.8 m/s and those with greater than 8.8 m/s. The obtained results showed that patients with acPWV equal or lower than 8.8 m/s survived longer than those with acPWV higher than 8.8 m/s. This acPWV value is near that proposed by other authors [35]. Taking into account that the CKD and ESKD are models for impaired VR and accelerated blood vessel aging [1,39], it is reasonable to expect a lower threshold for AS in comparison with the general population.

We found that higher acPWV values were associated with older age. On the other hand, PWV > 8.8 m/s was seen in 11 patients younger than 60.5 years. These individuals could be at a particularly high risk for a fatal outcome due to AS developing earlier than expected. Similar results were presented by Ferreira and coworkers using the acPWV cut-off of 12 m/s recommended for the general population [40]. Therefore, acPWV should preferentially be measured in younger dialysis patients in order to plan interventions prior to the development of irreversible changes in blood vessels.

Many authors point to an association of blood pressure and acPWV in the general population, as well as in CKD and dialysis patients [36,41]. Arterial stiffness directly influences blood pressure at higher systolic and pulse pressures, and vice versa, the elevated blood pressure related distension of arterial walls can increase AS [42]. This association is even stronger between acPWV and central systolic pressure, both measured in the aortic root. Systolic and diastolic aortic (central) blood pressures were stated to be better indicators of cardiovascular disease than brachial pressure [43], because aortic pressures are transmitted to vital organs, such as the heart, brain, and kidneys [43]. In agreement with this, our results showed that central systolic and brachial diastolic blood pressure were recognized as independent risk factors for a higher acPWV value. In order to detect silent hypertension and inter-dialysis and intra-dialysis blood pressure variabilities, some authors advise ambulatory blood pressure monitoring over 48 h [44,45]. At the same time, night-time systolic BP rather than day-time BPs was selected as a predictor of acPWV [45]. Since blood pressure largely influences acPWV, it might be expected that patients would live longer if acPWV decreased as a result of lower blood pressure. In practice, this only happens in patients with a decrease in acPWV parallel to the decline in blood pressure (i.e., pressure-sensitive AS), whereas non-survivors had a steady increase in acPWV despite a similar reduction in blood pressure (pressure-insensitive AS) [2,44], suggesting that pressure-insensitive AS is a major risk factor for mortality in hemodialysis patients. In addition, some authors have stated that given the dynamic fluid shifts among hemodialysis patients, changes in uremic milieu, and changes in bone mineral concentrations, measuring a static acPWV cannot be a predictor of dialysis patients’ mortality [44]. They proposed ambulatory acPWV monitoring over 48 h. However, the ability to monitor ambulatory PWV monitoring is limited in a number of research units, and in many research units, as in our one, a single measurement of PWV is carried out.

The present analysis showed that acPWV is positively associated with the pulse rate, which is a variable that we can influence. An association between acPWV and pulse rate has been described in the general population and in CKD patients but not in the dialysis population. A recent study showed minimal physiologically relevant changes of acPWV for small changes in pulse rate, but larger differences in pulse rate could be considered as contributing to significant differences in acPWV [46]. In addition, acPWV increased on average by 0.17 m/s per 10 bpm increase in pulse rate, independent of blood pressure changes, whereby there was a concurrent change in blood pressure with pulse rate [46]. The possible mechanism behind the influence of pulse rate on arterial stiffness is attributed to changes in smooth muscle tone in the large arteries, induced by altered sympathetic activity on the arterial wall [46].

Although statistically insignificant, higher iPTH, vitamin D, osteoprotegerin, and sclerostin on the one hand and lower FGF23 and Klotho on the other were found in our patients with acPWV > 8.8 m/s. The inclusion of these biomarkers in our analysis neither improved prediction of mortality of our patients nor selected them as independent variables for AS. Findings from previous studies are inconsistent. Some earlier studies showed that high levels of the biomarkers sclerostin, osteoprotegerin, FGF or FGF-KLOTHO axis, and/or vitamin D deficiency [46,47,48] correlated positively with AS, while others did not confirm the association of these biomarkers with AS in prevalent hemodialysis patients [49,50,51,52]. In comparison to the results published thus far, the higher values of FGF23 in the group with lower PWV (<8.8) found in our study is an unexpected finding. Once again, all variables from the database were analyzed, and we noticed a huge variation of data, but no explanation was found. These discrepant findings deserve further analyses of the role of FGF23 and other biomarkers in acPWV, either alone or in synergy with the other systemic and local factors. Having in mind all the aforementioned information, it can be speculated that AS might be a completed process in prevalent hemodialysis patients, and that various biomarkers are no longer involved in this. This speculation arose from Zoccali’s and London’s opinion on VC as smoke rather than fire in arterial disease. They underlined that, in the clinical setting, calcification follows inflammation of atherosclerosis and represents a secondary healing process [53]. Therefore, to maximize health benefits, the approach to vascular disease in CKD and dialysis patients should focus on prevention of arterial lesions by correcting traditional and non-traditional pro-atherogenic risk factors responsible for arterial injury, such as hyperphosphatemia and CKD mineral and bone disorders.

We failed to show any correlation between VC and AS. Moreover, significance of VC for patient survival was not confirmed in the multivariate analysis. However, the highest Adragao score of 8 was found in patients with acPWV over 8.8 m/s. In contrast to our analysis, others reported a close relationship between VC and AS, indicating that more calcified arteries lose their elasticity [54]. Furthermore, it was demonstrated that both increased AS and cardiovascular calcifications were major predictors of all-cause and CV mortality in renal patients [2,5,6]. An explanation for this disparity may lie in the different sites of VC assessment and the methods used for diagnosis of VC. We estimated the Adragao score for iliac, femoral, radial, and digital arteries in plain radiographic films. In addition to these blood vessels, others evaluated calcifications on arteriovenous fistulae, heart valves, or carotid or coronary blood vessels using the same method or computed tomography. Some data indicate that the association of VC and acPWV depends on where VC was examined. Thus, there was association of acPWV and abdominal aorta calcification, a weaker association of calcification with coronary arteries, and thoracic aorta calcification but no association with valvular calcification [55].

There are several limitations to this study that need to be noted. First, the number of patients enrolled was small and our findings need to be confirmed in a larger cohort. Second, a longer follow-up may have led to more events. Finally, this was an observational dataset, which cannot account for the presence of residual confounding factors, and causality can only be implied.

## 5. Conclusions

Among prevalent patients treated with hemodialysis, we observed that AS, measured with acPWV, independently predicted all-cause mortality. A value for acPWV over 8.8 m/s was associated with a greater risk of patient mortality, especially in those younger than 60.5 years. Therefore, we can advise the measurement of acPWV preferentially in younger dialysis patients for prognosis, as well as intervention planning before the development of irreversible changes in blood vessels. In addition, measuring central systolic blood pressure seems to be useful for monitoring AS in prevalent hemodialysis patients.

## Figures and Tables

**Figure 1 medicina-56-00435-f001:**
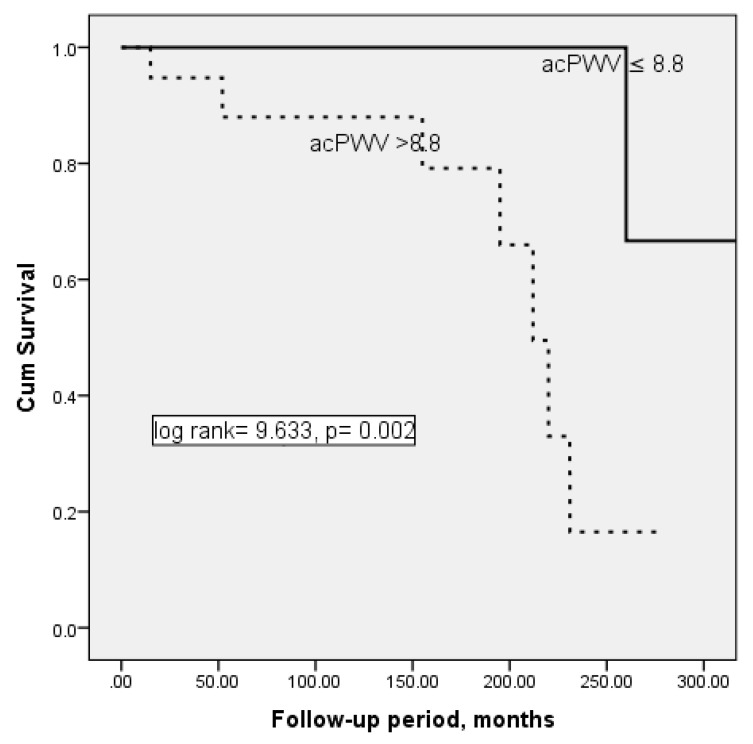
Survival plots for ankle carotid pulse wave velocity (acPWV) groups, i.e., acPWV ≤ 8.8 and acPWV > 8.8 (Kaplan–Meier analysis).

**Table 1 medicina-56-00435-t001:** Basal characteristics, presence of co-morbidities, and treatment of the patients divided by pulse wave velocity median value.

Variables	PWV ≤ 8.8 m/s	PWV > 8.8 m/s	*p*
No of patients	36	36	
Dialysis vintage, months	123.34 ± 86.23	110.47 ± 81.48	0.520
Age, years	50.71 ± 12.26	61.89 ± 11.96	0.000
Gender, males (%)	14	20	0.119
BMI, kg/m^2^	24.03 ± 4.13	25.31 ± 3.72	0.190
kT/V Primary kidney disease: GN Nephroangiosclerosis APCKD Calculosis BEN Others Treatment: yes, (%) Vitamin D Phosphate binders Calcium-based ESA Anti-hypertensive drugs	1.27 ± 0.27 6 5 5 10 2 8 12 33 16 24	1.25 ± 0.16 8 9 3 2 10 4 12 34 15 25	0.938 0.767 0.372 0.710 0.024 0.024 0.599 0.500 0.192 0.500
Co-morbidities, yes, (%)			
Hypertension	26	20	0.110
CVD	7	12	0.142
CVI	4	2	0.337

Categorical variables are presented as absolute values; continuous normally distributed variables are presented as mean ± SD; continuous skewed variables are presented as median (interquartile range (IQR)). BMI: body mass index, kT/V: single pool urea kinetic, GN: glomerular disease, APCKD: adult polycystic kidney disease, BEN: Balkan endemic nephropathy, ESA: erythropoietin-stimulating agents, CVD: cardiovascular diseases (pre-study period: heart failure, previous myocardial infarction, aorto-coronary bypass surgery, peripheral vascular disease), CVI: cerebrovascular insult.

**Table 2 medicina-56-00435-t002:** Cardio-vascular parameters in patient groups.

Variables	PWV ≤ 8.8 m/s	PWV > 8.8 m/s	*p*
Adragao score Number of patients *: 0 1–3 >4	2.0 (5.25) 9 8 9	1.0 (4.0) 12 7 8	0.452
PWV, m/s	7.44 ± 0.83	11.22 ± 2.03	0.000
ABI	1.18 ± 0.13	1.15 ± 0.20	0.407
MAP, mmHg	87.77 ± 12.28	102.54 ± 18.59	0.002
cSBP, mmHg bSBP, mmHg	124.37 ± 28.37 108.18 ± 36.98	145.10 ± 28.37 131.64 ± 36.93	0.005 0.014
bD BP, mmHg	62.88 ± 20.71	78.06 ± 16.03	0.002
Heart rate/min	75.64 ± 13.04	82.48 ± 12.08	0.033
Pulse pressure	53.67 ± 19.84	63.81 ± 18.84	0.041

Continuous normally distributed variables are presented as mean ± SD; continuous skewed variables are presented as median (IQR). cSBP: central systolic blood pressure, bSBP: brachial systolic blood pressure, bDBP: brachial diastolic blood pressure. * The vascular calcifications (by Adragao score) were estimated in 53 patients.

**Table 3 medicina-56-00435-t003:** Basal laboratory data.

Variables	PWV ≤ 8.8	PWV > 8.8	*p*
S-urea, mmol/L	19.65 ± 5.70	23.27 ± 6.85	0.017
S-creatinine, umol/L	825.53 ± 206.32	853.81 ± 210.30	0.567
S-urate, umol/L	345.69 ± 78.69	364.75 ± 97.33	0.364
Hemoglobin, g/L	109.33 ± 11.90	107.78 ± 11.59	0.539
Leucocyte no., ×10^9^/L	5.91 ± 1.62	6.24 ± 2.68	0.532
Platelet count, ×10^3^/µL	196.44 ± 59.05	190.31 ± 64.34	0.675
S-Soduim, mmol/L	137.94 ± 2.47	137.83 ± 3.32	0.872
S-Calcium, mmol/L	2.18 ± 0.37	2.19 ± 0.17	0.855
S-Phosphate, mmol/L	1.69 ± 0.43	1.67 ± 0.50	0.794
Alkaline phosphatase, IU/L	83.11 ± 41.55	97.22 ± 39.36	0.049
iPTH, pg/mL	45 (179)	142 (269)	0.168
1,25(OH)2D, ng/mL	36.43 ± 20.73	332.95 ± 15.11	0.427
FGF 23, pg/mL	1500 (830)	1106 (1168)	0.574
Klotho, ng/mL	0.039 (0.07)	0.034 (0.015)	0.335
Osteoprotegerin	0.089 (0.430)	0.14 (1.70)	0.814
S-Magnesium, mmol/L	1.23 ± 0.20	1.17 ± 0.19	0.207
Cholesterol, mmol/L	4.34 ± 1.05	4.85 ± 1.35	0.144
Triglycerides, mmol/L	1.60 (1.60)	1.7 (1.7)	0.542

Continuous normally distributed variables are presented as mean ± SD; continuous skewed variables are presented as median (IQR).

**Table 4 medicina-56-00435-t004:** Predictors of mortality selected by Cox regression analysis.

	B	Significance	Exp (B)	95.0% CI for Exp (B)
acPWV (median)	2.688	0.023	14.696	1.450 148.91
Klotho	1.019	0.660	2.770	0.030 258.38
Hypertension	−1.401	0.127	0.246	0.041 1.493
Age	0.012	0.799	1.012	0.925 1.107

**Table 5 medicina-56-00435-t005:** Independent variables analysis associated with pulse wave velocity (PWV) selected by logistic regression.

		B	Significance	Exp (B)	95% CI for EXP (B)
**Model 1 ^*^**	Age	0.082	0.007	1.085	1.023 1.152
	cSBP	0.025	0.038	1.025	1.001 1.050
	Heart rate	0.067	0.016	1.069	1.012 1.128
	Constant	−14.062	0.000	0.000	
**Model 2 ****	Age	0.100	0.003	1.105	1.035 1.181
	Heart rate	0.071	0.017	1.074	1.013 1.138
	bDBP	0.054	0.006	1.055	1.015 1.096
	Constant	−16.396	0.000	0.000	

Dependent variable was acPWV used as binary variable—acPWV ≤ 8.8 was coded as 0, and acPWV > 8.8 was coded as 1 (according to median acPWV). * Model 1 included all variables listed in Table 1, Table 2 and Table 3, but brachial systolic and diastolic blood pressure; ** Model 2 included all variables listed in Table 1, Table 2 and Table 3, but central systolic blood pressure

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
