# Peer review of "Significance of acPWV for Survival of Hemodialysis Patients"

_medicina, 2020, doi:10.3390/medicina56090435_

Round 1

Reviewer 1 Report

 Over all well written paper, however  could consider following recommendations to increase quality of the paper.

1)acPWV  is checked in current study by two different personal and mean value is taken into consideration. However given the dynamic fluid shifts among hemodialysis patients, changes in uremic milieu and changes in bone mineral concentrations, measuring a static acPWV cannot be correlated with mortality. It could be subjected to variations dynamically in ESRD patients.  Pts on HD will have inter dialysis and intra dialysis blood pressure variabilities

2) Ambulatory blood pressure monitoring is more correlated with blood pressures among ESRD patients than single measurements taken once before HD sessions

3) This paper did not mention if they excluded patients with severe uncontrolled blood pressures.

4) Mean KT/V values in comparison in two different groups are not mentioned

5)Table 3 demonstrates High FGF -23 levels among pts with ac PVW < 8.8, however we know from previous literature , high FGF -23 levels associated with increased LVH and further contributing to increased adverse effects which  is a  contradictory finding in lower acPVW group

Author Response

1)acPWV  is checked in current study by two different personal and mean value is taken into consideration. However given the dynamic fluid shifts among hemodialysis patients, changes in uremic milieu and changes in bone mineral concentrations, measuring a static acPWV cannot be correlated with mortality. It could be subjected to variations dynamically in ESRD patients.  Pts on HD will have inter dialysis and intra dialysis blood pressure variabilities

Discussion, page 9, at the end of second paragraph- the importance of ambulatory PWV monitoring is stated and also the limited possibility to do it in many research units

2) Ambulatory blood pressure monitoring is more correlated with blood pressures among ESRD patients than single measurements taken once before HD sessions

Discussion, Page 9, second paragraph, sixth sentence- discussion on significance of ABPM. The new reference is added (Li X et. al). As a result, the reference numbers have been moved accordingly

3) This paper did not mention if they excluded patients with severe uncontrolled blood pressures.

Patients with uncontrolled blood pressure were excluded from the study and this was added in section 2.Materials and methods, 2.1. Study groups, as the 6th criterion for inclusion / exclusion of patients from the study

4) Mean KT/V values in comparison in two different groups are not mentioned

The kT / V values were added for both groups in Table 1

5)Table 3 demonstrates High FGF -23 levels among pts with ac PVW < 8.8, however we know from previous literature , high FGF -23 levels associated with increased LVH and further contributing to increased adverse effects which  is a  contradictory finding in lower acPVW group.

Page 10, first paragraph and second sentence- a comment on this remark is given

Reviewer 2 Report

The authors present a well designed observational, non-interventional study regarding the use of PWv in hemodialysis patients.

The authors should explain in the Materials and methods section the exact timeframe when the procedures (biochemical, calcification and vascular assessments) were performed- probably on admission after signing the informed consent, but it should be clearly stated, because the patients have been followed up for almost 5 years. There is also no information regarding the medication after inclusion.

The authors should explain the lower cut-off value (median value) of PWV obtained compared to the cut-off values presented by other authors in different papers cited.

Because of the rather small number of subjects, a conclusion such as : "the control of blood pressure seems to be more important then measurement of bone biomarkers" is probably to radical and should be changed accordingly.

Minor suggestions:

Page 1, line 31 Please replace “probability of dying” with mortality

Page 2, line 73 The authors could rephrase the last sentence of the introduction as follows: “…aim to determine the significance of AS for survival of… , and the potential independent association of AS with…”

Author Response

1. The authors should explain in the Materials and methods section the exact timeframe when the procedures (biochemical, calcification and vascular assessments) were performed- probably on admission after signing the informed consent, but it should be clearly stated, because the patients have been followed up for almost 5 years. There is also no information regarding the medication after inclusion.

Additional explanation of the timeframe when the procedures were done is added at the end of second paragraph of section 2.1.

2. The authors should explain the lower cut-off value (median value) of PWV obtained compared to the cut-off values presented by other authors in different papers cited.

The possible explanation is given in Discussion, third paragraph, fifth sentence, line 273

3. Because of the rather small number of subjects, a conclusion such as : "the control of blood pressure seems to be more important then measurement of bone biomarkers" is probably to radical and should be changed accordingly.

The last sentence in Abstract and Conclusion is changed.

Minor suggestions:

Page 1, line 31 Please replace “probability of dying” with mortality

Corrected

Page 2, line 73 The authors could rephrase the last sentence of the introduction as follows: “…aim to determine the significance of AS for survival of… , and the potential independent association of AS with…”

Corrected

Round 2

Reviewer 1 Report

Significant improvement has been made to the paper.

No further recommendations.